# Hybrid Lead-Halide Polyelectrolytes as Interfacial Electron Extraction Layers in Inverted Organic Solar Cells

**DOI:** 10.3390/polym12040743

**Published:** 2020-03-27

**Authors:** Jin Hee Lee, Yu Jung Park, Jung Hwa Seo, Bright Walker

**Affiliations:** 1Department of Materials Physics, Dong-A University, Busan 49315, Korea; dljh82@gmail.com (J.H.L.); j6768kr@gmail.com (Y.J.P.); 2Department of Chemistry, Kyung Hee University, Seoul 02447, Korea

**Keywords:** electron extraction layer, nonconjugated polyelectrolytes, lead-halide

## Abstract

A series of lead-halide based hybrid polyelectrolytes was prepared and used as interfacial layers in organic solar cells (OSCs) to explore their effect on the energy band structures and performance of OSCs. Nonconjugated polyelectrolytes based on ethoxylated polyethylenimine (PEIE) complexed with PbX_2_ (I, Br, and Cl) were prepared as polymeric analogs of the perovskite semiconductors CH_3_NH_3_PbX_3_. The organic/inorganic hybrid composites were deposited onto Indium tin oxide (ITO) substrates by solution processing, and ultraviolet photoelectron spectroscopy (UPS) measurements confirmed that the polyelectrolytes allowed the work function of the substrates to be controlled. In addition, X-ray photoelectron spectroscopy (XPS) results showed that Pb(II) halide complexes were present in the thin film and that the Pb halide species did not bond covalently with the cationic polymer and confirmed the absence of additional chemical bonds. The composite ratio of organic and inorganic materials was optimized to improve the performance of OSCs. When PbBr_2_ was complexed with the PEIE material, the efficiency increased up to 3.567% via improvements in open circuit voltage and fill factor from the control device (0.3%). These results demonstrate that lead-halide based polyelectrolytes constitute hybrid interfacial layers which provide a novel route to control device characteristics via variation of the lead halide composition.

## 1. Introduction

Organic solar cells (OSCs) offer many advantages as an alternative energy technology. Their light weight, simplicity of processing, and flexible nature have made them the focus of an intense, worldwide research effort [1,2,3,4]. Many aspects of OSCs have developed over the past two decades, and a detailed understanding of how various elements of OSCs affect the device characteristics has developed [5,6]. The basic OSC structure consists of an active layer between a transparent, conductive anode (usually indium tin oxide, ITO) and a metal cathode. The organic active layer consists of an electron donor/electron acceptor pair which is able to undergo photo-induced electron transfer upon exposure to light. The first iterations of OSCs involved active layers consisting of 2-dimensional donor/acceptor bilayers [7], but the most effective active layer structure which has emerged is the bulk heterojunction (BHJ), which consists of a 3-dimensionsal blend of donor and acceptor materials which offers increased efficiency due to the large donor/acceptor interfacial area [8,9]. The BHJ structure usually consists of a blend of polymer donor and fullerene acceptor based on C_60_ or C_70_. In this work, we have employed a bulk heterojunction consisting of poly(3-hexylthiophene) (P3HT) as a donor, and [6,6]-phenyl-C_61_-butyric acid methyl ester (PC_61_BM) as an acceptor. This system constitutes the most thoroughly studied OSC system and makes a suitable platform for studying the effects of new interfacial materials [10].

Conventional OSC structures usually use aluminum as a metal cathode, because it possesses a relatively low work function (ϕ) and high reflectivity and can be easily thermally evaporated. Although aluminum is fairly stable relative to other low-ϕ metals like Ca or Mg, it still has the disadvantage of being easily oxidized and having low stability relative to higher-ϕ metals like Au or Ag [11]. To overcome this stability problem, OSCs with the inverted structure have been successfully developed and demonstrated to offer higher stability [12]. In an OSC device with the conventional structure, a p-type substrate is used as the anode and low-ϕ aluminum is used as the cathode, resulting in a device with moderate stability [13], while in inverted OSC structures, an n-type substrate is used with high-ϕ metal anodes like Au or Ag deposited on top, which offers improved stability because Au and Ag are not easily oxidized. OSCs with the inverted structure exhibit high stability compared to the conventional structure, making commercial production more practical [14].

While conventional OSC devices use ITO substrates coated with poly(3,4-ethylenedioxythiophene)-poly(styrenesulfonate) PEDOT:PSS to create a p-type anode, an n-type electron transport layer (ETL) must be used to facilitate electron collection so that the substrate functions effectively as a cathode in inverted devices. The most commonly used ETLs in this context include zinc oxide (ZnO) and titanium oxide (TiOx) [15,16,17]. A significant disadvantage of using such inorganic metal oxides is that high-temperature annealing/sintering treatment is usually required [18]. Such high-temperature processing negates some of the main advantages of OSC devices such as low-temperature processability and flexibility [19].

With the aim of producing stable, inverted solar cells via a simple solution processing method, we sought polymer-based interfacial materials which could be used without high-temperature processing [20]. Polymer-based interfacial materials have been shown to form dipoles on ITO electrodes which allows ϕ to be adjusted [21]. Conjugated polyelectrolytes (CPEs) have been used extensively to lower the work function of electrodes in organic semiconducting devices [22,23,24], and nonconjugated polyelectrolytes (NPEs) [25,26] are representative types of polymers which may be used for this purpose [5,27,28]. CPEs are based on repeating aromatic backbones but have the disadvantage that the conjugated, aromatic backbone is costly to synthesize and may introduce extraneous electronic effects, whereas NPEs can be synthesized simply and inexpensively, and allow the effects of ionic groups to be investigated without interference from a conjugated backbone [29,30,31]. Therefore, NPEs constitute an ideal platform to study the effects of ionic materials on the energy bands in OSCs. NPEs are easy to make, easy to deposit on substrates at room temperature, and allow the influence of their ionic constituents to be evaluated without interference from other conjugated organic moieties [29,30,31].

Perovskite solar cells and hybrid solar cells based on mixed organic and inorganic components have achieved remarkably high efficiencies in the past two decades [32,33]. Lead halide perovskite (LHP) semiconductors, which are known to have remarkably good electronic properties such as high-charge carrier mobilities [34], low-charge carrier recombination rates, and large carrier diffusion lengths [35], as well as tunable energy band structures [36], are based on combinations of Pb(II) halide salts and halides of organic cations such as methylammonium iodide (MAI) or formamidinium iodide. This type of semiconductor can be formed by simply mixing two components in a suitable solvent and allowing the solvent to evaporate.

Some of the NPEs which have been used as ETLs in OSCs bear resemblance to the organic cations used in perovskites. For example, the NPE based on polyethylenimine (PEI) protonated with Hydrogen Iodide (HI) contains the repeat unit (CH_2_CH_2_NH_2_^+^I^−^)_n_, which is chemically similar to the organic cation MAI (CH_3_NH_3_^+^I^−^). In analogy to the preparation of LHP semiconductors, the incorporation of Pb(II) halides in the NPE may lead to polymeric, perovskite-like hybrid materials as illustrated in Figure 1. Such lead halide polyelectrolytes might be expected to yield materials with functional electronic properties like LHPs. Notably, the Pb(II) halide component may provide a new avenue to control the electronic properties and function of NPE interfacial materials. To the best of our knowledge, such polyelectrolytes based on complex lead halide anions have not yet been reported. In this work, we seek to explore this possibility by preparing a series of NPE:PbX_2_ composites and investigating their influence on the electronic structure and characteristics of OSCs.

## 2. Experimental

Hybrid lead halide polyelectrolytes were prepared as follows. A solution of ethoxylated PEI (PEIE) in deionized water was made at solids concentrations of ~0.25 g/mL. Aqueous solutions of acid (HI, HBr, and HCl) were slowly added to the polymer solutions until test aliquots (5 μL of polymer solution diluted in 1 mL water) reached a pH in the range of 5.0 to 5.5 (Sigma-Aldrich, Seoul, Korea). The volume of acid added was recorded to calculate the fraction of protonated repeat groups in the polymer. Based on the amount of acid added, the molar fraction of repeat units bearing a positive charge and halide anion was calculated to be 49.3%, 40.6%, and 40.6% for PEIE titrated with HCl, HBr, and HI, respectively. The polymer solutions were first precipitated into isopropanol yielding milky suspensions. The suspensions were centrifuged to isolate the polymers as dense, viscous liquids which still contained a significant amount of water. In order to remove the water, the polymers were subsequently re-suspended in anhydrous methanol (without complete dissolution), and precipitated again into anhydrous ethanol. The polymers were finally rinsed with ether and dried under vacuum to yield resinous solids (Sigma-Aldrich, Seoul, Korea). The ionic polymers are referred to as PEIEH^+^Cl^−^, PEIEH^+^Br^−^ and PEIEH^+^I^−^ for polymers titrated with HCl, HBr, and HI, respectively. In an adaptation of procedures used to prepare CH_3_NH_3_PbI_3_ perovskite solutions, polyelectrolyte-metal complex solutions were prepared by dissolving anhydrous Pb halides in dimethylsulfoxide (DMSO) and mixing them with solutions of the polyelectrolytes a 1:1 mole ratio of X^−^ in the polyelectrolyte to PbX_2_ to form PEIEH^+^PbX_3_^−^, then diluting with PEIE in 2-methoxyethanol using a micropipette to achieve various ratios of polymer to PbX_2_.

Inverted OSC devices were fabricated using the following procedure. First, ITO coated glass substrates were cleaned with detergent, then ultrasonicated in distilled water and isopropyl alcohol, then dried in an oven at 100 °C. A mixture of PEIEH^+^X^−^ and PbX_2_ (X = I, Br, and Cl) in dimethyl sulfoxide/2-methoxyethanol was filtered through a 0.45 μm polytetrafluoroethylene filter and spin coated at 2000 rpm for 40 s air (All the chemicals and solvent used in this work were purchased from Sigma-Aldrich, Seoul, Korea). Organic/inorganic hybrid composites layers were then dried under vacuum in an anti-chamber. Subsequently, substrates were transferred into a nitrogen filled glove box. Active layers were deposited by spin coating a composite solution of P3HT:PC_61_BM (1:0.8) in o-dichlorobenzene (o-DCB) at 700 rpm for 60 s on top of the substrate to obtain a BHJ film with thickness of approximately 200 nm. Samples were then brought under high vacuum (about 10^−7^ Torr), and MoO_x_ (≈ 5.0 nm), followed by Ag electrodes (100 nm), was deposited on top of the BHJ layer by thermal evaporation (All the chemicals and solvent used in this work were purchased from Sigma-Aldrich, Seoul, Korea). Devices had an active area of 0.124 cm^2^, defined by an aperture. Then, 16–20 devices were fabricated for each condition. Current density-voltage (J-V) measurements were collected using a Keithley 2635 source measure unit (Tektronix, Seoul, Korea) and carried out inside a nitrogen filled glove-box using a high quality optical fiber to guide the light from a xenon arc lamp (HS Technology, Seoul, Korea) to the solar cell device. The solar cell devices were illuminated with an intensity of 100 mW/cm^2^ as calibrated using a standard silicon reference cell.

## 3. Results and Discussions

Figure 1 shows the structures of the materials and the device test structure. As discussed previously, the organic cations typical of perovskite semiconductors were substituted with polymeric cations based on the PEIE [25,37] structure to yield a polymeric lead halide composite. Thin films of these NPE materials were deposited onto ITO substrates via spin coating.

Figure 2 shows J-V characteristics of OSCs with the organic/inorganic hybrid composites with I^−^, Br^−^, and Cl^−^ anions. The photovoltaic parameters are summarized in Table 1. The J-V curve and device parameters of the device with PEIE ETL as a refence cell was shown in Appendix A and Appendix A, respectively. For PEIE:PbCl_x_ interfacial layers (Figure 1a), the best performance included a relatively high open-circuit voltage (V_OC_) of 0.622 V and a high fill factor (FF) of 65.65% at an ethoxylated polyethyleneimine - lead chloride complex (PEIE:PbCl_x_) ratio of 97:3. Higher loading of the PbCl_x_ component led to decreased short-circuit current (J_SC_), V_OC_, and FF. Figure 1b shows the effects of PbBr_2_-based ETLs on device characteristics. In the case of PbBr_2_, a FF of up to 69.06% and power conversion efficiency (PCE) of up to 3.567% were observed for PbBr_2_ concentration of 1%. Increasing the PbBr_2_ content to 2% and beyond led to a gradual decreases in both FF and V_OC_. Figure 1c shows data corresponding to PbI_2_-based devices. The average V_OC_ for devices processed with PbI_2_ was lower than the other halides, which led to generally lower PCE values. As with Cl^−^, a concentration of 3% yielded the greatest FF value (66.63%) although the highest PCE value (3.052%) was observed for the 1% concentration due to a slightly larger V_OC_ (0.602 V) obtained at this concentration. External quantum efficiency (EQE) results for the devices with PEIE:PbI_2_ ETLs were conducted to measure the spectral J_SC_ and were shown in Appendix A (PV Ins., Point Roberts, WA98281, USA). Overall, PbBr_2_ yielded the best device characteristics with the highest overall J_SC_, FF, and PCE values among the three halides. The optimal processing concentration of 1% yielded a champion device with a J_SC_ of 8.278 mA/cm^2^, a V_OC_ of 0.624 V, a FF of 69.06%, and a PCE of 3.567%.

X-ray photoelectron spectroscopy (XPS) was performed to investigate the material composition and bonding states for the different elements in the ETL films. The intensities of N 1s XPS spectra (Figure 3a–c) for PEIE:PbCl_2_, ethoxylated polyethyleneimine - lead bromide complex (PEIE:PbBr_2_), and ethoxylated polyethyleneimine - lead iodide complex (PEIE:PbI_2_) films were attenuated slightly as the proportion of inorganic lead halide increased. The binding energy of the N 1s peaks was consistent with nitrogen in a tertiary organic amine bonding state, which was expected to be the predominant bonding state in the PEIE backbone [38]. Previous NMR studies of the materials confirmed the degree of functionalization in the backbone (nitrogen quaternization) of this series of polyelectrolytes [39]. Figure 3d–f corresponds to photoelectron emission bands of Pb at 138.7 eV from all films, consistent with the +2 oxidation state in this metal. The intensities increased as the proportion of inorganic salts increased. Although the sensitivity for the elements Cl and Br was low, PbCl_2_, PbBr_2_, and PbI_2_ features appeared at 198 eV for Cl 2p, 68.3 eV for Br 3d, and 618.5 eV for I 3d, respectively, consistent with the elements existing as halide anions in their anionic bonding states.

Figure 4 shows the secondary edge region of ultraviolet photoelectron spectra (UPS) for each material. The cutoff value changes with each halide as well as with the ratio of PbX_2_ to polymer. This data confirms that the energy bands are influenced by each component in the ETL. The ϕ can be calculated using this equation using UPS.
(1)$$\phi =h\nu \;-E_{k}$$

Here, h is the Planck constant, ν is the frequency, and E_k_ is the maximum kinetic energy from the photoelectron. Comparing the ϕ before and after addition of inorganic materials, we see that chloride caused a change in ϕ of 0.06 eV, bromide shifted ϕ by 0.08 eV, and iodide had the largest effect on ϕ at 0.1 eV. The shift of the cutoff energies indicates the magnitude of the interfacial dipole (Δ), which is equal to the changes in ϕ. The exact magnitude of the Δ is likely a result of an interplay between the intrinsic electronic structure of the polyelectrolytes at the metal/polymer interfaces. Although the highest Δ value was observed for the PEIE:PbI_2_ film, the best efficiency was obtained from the device with the PEIE:PbBr_2_ ETL. To investigate this point, topographic studies were performed via atomic force microscopy (AFM, Bruker Nano Inc., Santa Barbara, USA), as shown in Appendix A. The PEIE:PbCl_2_ films exhibited very rough morphology, with relatively large roughness, while the PEIE:PbI_2_ films on ITO showed large granules on the surface. However, the PEIE:PbBr_2_ films exhibited the most smooth and even surface among the polyelectrolytes, resulting in the highest efficiency. In Appendix A, the water contact angles differed depending on the type of halide. Br-based films showed relatively high contact angles at all proportions, indicating a relatively hydrophobic surface. The hydrophobic surface resulted in good wetting and compatibility when the active layer was deposited on the Br films, while the other, more hydrophilic surfaces had poorer wetting between the polyelectrolyte and the nonpolar active layer solution.

## 4. Conclusions

A series of perovskite-like organic/inorganic hybrid polyelectrolytes were prepared, and their influence on solar cell devices was characterized. XPS data confirmed the presence of Pb(II) and halides consistent with the chemical composition of each material, and indicated that the Pb(II) species did not participate in bonding with the cationic organic polymer but remained coordinated to halogen atoms. The influence of hybrid, lead halide electrolytes on the energy band structure of BHJ and ITO film was characterized. UPS measurements confirmed that the lead halide electrolytes were able to modulate the ϕ value of ITO, and allowed the ϕ of the surface to be reduced to values appropriate for n-type carrier extraction. Inverted OSC devices were fabricated using a BHJ structure with a P3HT:PC_61_BM active layer and high-ϕ, Ag anodes. Reduced work functions confirmed that the new electrolytes functioned as effective ETLs in these devices. As the Pb(II) halide content increased, the J_SC_ and FF were increased for concentrations in the range of 1% to 3% but deteriorated for high concentrations of Pb(II). PbBr_2_ yielded the best performance, including increases in FF and V_OC_, while PbI_2_ was the least effective due to a small decrease in V_OC_. This report introduces a new perovskite-like class of hybrid, lead-halide polyelectrolyte ETL layer for application in OSCs and confirms that the J_SC_ and FF can be modulated and improved even for very small concentrations of Pb(II).

## Figures and Tables

**Figure 1 polymers-12-00743-f001:**
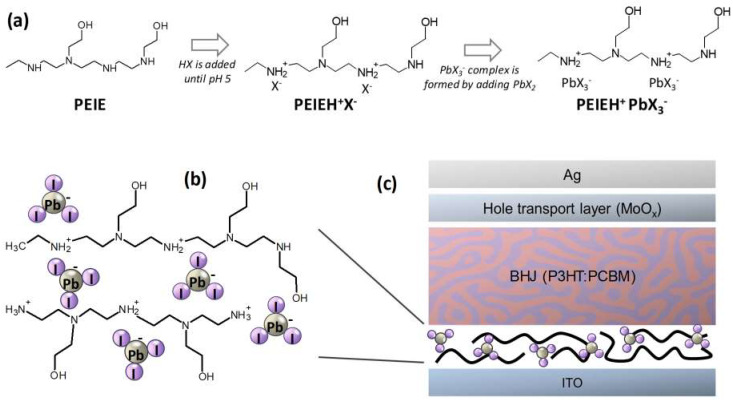
Schematic illustrations showing (**a**) the preparation of a hybrid lead halide polyelectrolyte. (**b**) Conceptual diagram of the chemical structure of the ethoxylated polyethyleneiminium lead triiodide (PEIEH^+^PbI_3_^−^) polyelectrolyte and (**c**) device diagram showing the application of ethoxylated polyethyleneimine-lead halide complex (PEIE:PbX_2_) polyelectrolytes as a solution-processible hybrid electron transport layer in an inverted OSC.

**Figure 2 polymers-12-00743-f002:**
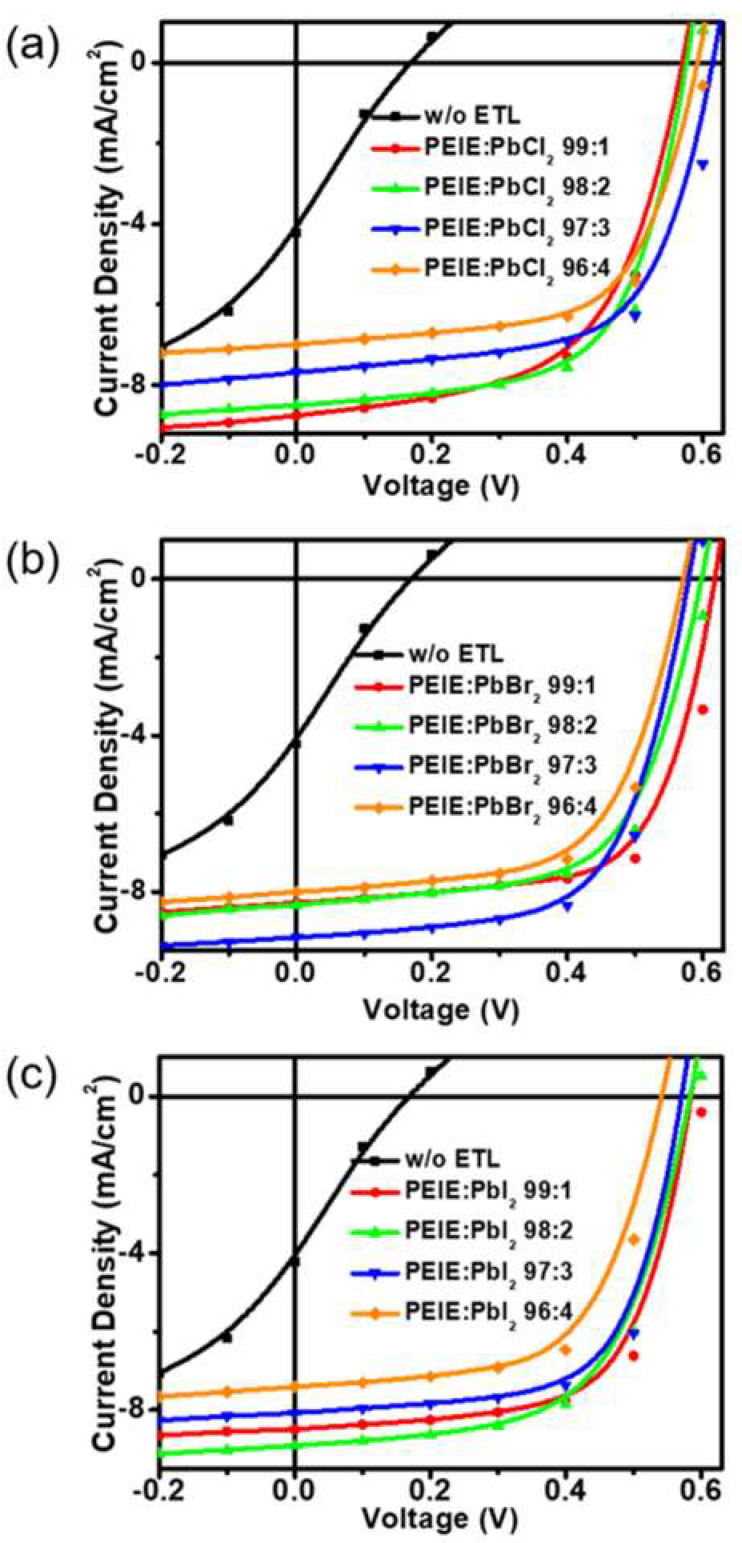
J-V characteristics under illumination of inverted P3HT:PC_61_BM solar cells with various electron transport layers, (**a**) PEIE:PbCl_2_, (**b**) PEIE:PbBr_2_, and (**c**) PEIE:PbI_2_, as a function of PEIE and PbX_2_ ratio.

**Figure 3 polymers-12-00743-f003:**
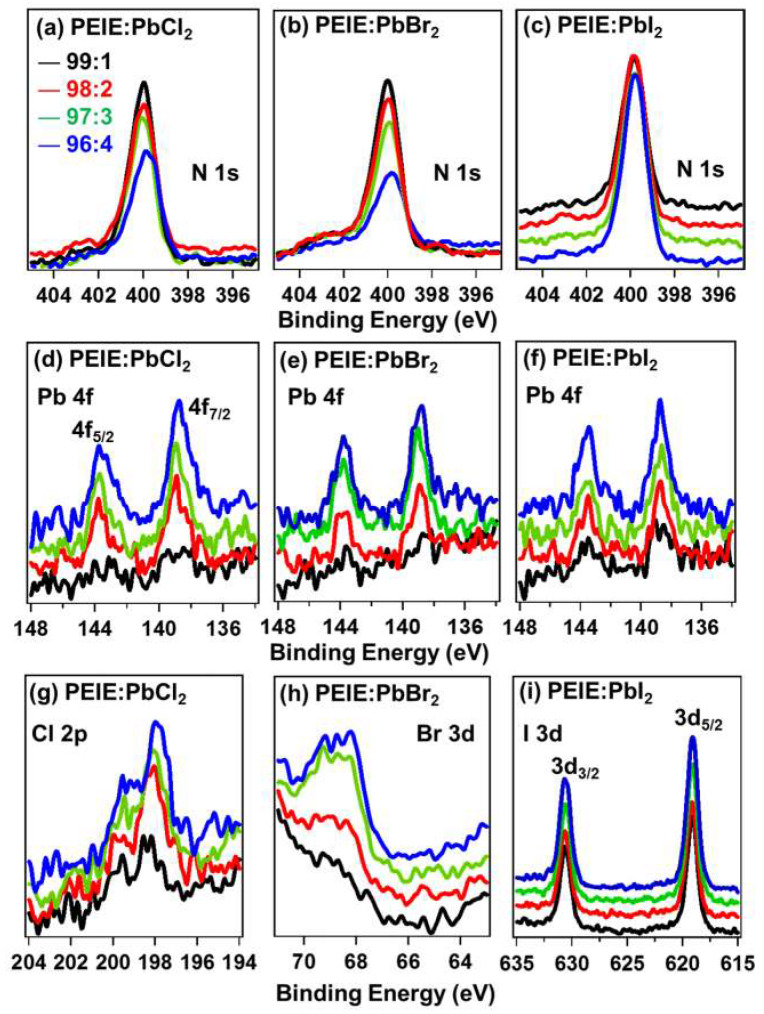
Quantitative analysis of XPS results. Nitrogen 1s emission levels for (**a**) PEIE:PbCl_2_, (**b**) PEIE:PbBr_2_, and (**c**) PEIE:PbI_2_ films, Lead 4f emission levels for (**d**) PEIE:PbCl_2_, (**e**) PEIE:PbBr_2_, and (**f**) PEIE:PbI_2_ films and Chlorine 2p emission levels for (**g**) PEIE:PbCl_2_, Bromine 3d emission levels for (**h**) PEIE:PbBr_2_, and Iodine 3d emission levels for (**i**) PEIE:PbI_2_ films as a function of the organic and inorganic composite ratio.

**Figure 4 polymers-12-00743-f004:**
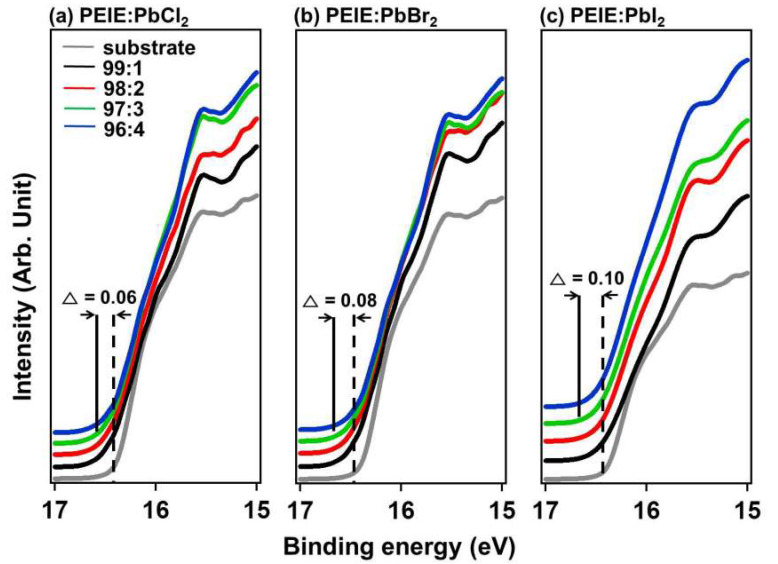
Secondary electron cutoff ultraviolet photoelectron spectroscopy (UPS) spectra for (**a**) PEIE:PbCl_2_, (**b**) PEIE:PbBr_2_, and (**c**) PEIE:PbI_2_ films as a function of the organic and inorganic composite ratio.

**Table 1 polymers-12-00743-t001:** Device characteristics of inverted poly(3-hexylthiophene):[6,6]-phenyl-C_61_-butyric acid methyl ester (P3HT:PC_61_BM) solar cells.

ETLs	J_SC_ (mA/cm^2^)	V_OC_ (V)	FF (%)	PCE (%)	
				Average	Best
without	5.674	0.150	29.40	0.250	0.300
PEIE	8.865	0.544	59.98	2.687	2.894
PEIE:PbCl_2_					
99:1	8.755	0.590	56.47	2.784	2.917
98:2	8.508	0.596	60.53	2.589	3.069
97:3	7.690	0.622	65.65	2.871	3.140
96:4	6.990	0.605	64.19	2.709	2.715
PEIE:PbBr_2_					
99:1	8.278	0.624	69.06	2.991	3.567
98:2	8.337	0.608	63.26	2.922	3.207
97:3	9.159	0.593	61.40	3.097	3.335
96:4	7.991	0.587	61.07	2.263	2.865
PEIE:PbI_2_					
99:1	8.499	0.602	64.70	3.052	3.310
98:2	8.916	0.596	59.21	2.821	3.146
97:3	8.135	0.599	66.63	2.839	3.247
96:4	7.378	0.545	64.16	1.854	2.579

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
