# Peer review of "Hybrid Lead-Halide Polyelectrolytes as Interfacial Electron Extraction Layers in Inverted Organic Solar Cells"

_polymers, 2020, doi:10.3390/polym12040743_

Round 1
Reviewer 1 Report
Lee and coworkers reported the synthesis of a series of lead-halide based hybrid polyelectrolytes and explored their effect on the energy band structures and performance of P3HT-based organic solar cells (OSCs).
PbBr2 yielded the best device characteristics with the highest overall JSC, FF and PCE values among the three halides. The photovoltaic performance of the present OSCs is relative poor. While, actually, P3HT is still the main donor material for the fabrication of large area PSCs, because it can be synthesized in large scale with relatively low cost.
Overall, the tendency of the device performance observed is not surprising but reasonable from the viewpoint of the current understanding of OPVs. I recommend this manuscript can be accepted for publication in Polymers after major revisions below.
(1) OSCs with high conductivity polyelectrolytes as the electron transport layer are good for electron collection. Therefore, the conductivity of the solid NPE films before and after PbX2 treatment need be measured to better understand the photovoltaic performance.
(2) The authors should give a detailed discussion on the possible reasons why OSC with PbBr2 modified PEIE as the EML exhibited the best efficiency.
(3) The authors didn't mention whether the synthesized hybrids are novel or there were previous studies on the identical structures. If there are other similar studies, they should be cited.
Reviewer 2 Report
This work shows a hybrid lead-halide polyelectrolytes as interfacial electron extraction layers for organic solar cells. Before it can be accepted for publication, the following questions should be answered.
- The performance of PEIE-only cells should be listed as the reference devices.
- The characterization results of UPS, AFM, and water contact angles results should be further discussed to support the different performance of PEIE-PbX2 devices.
- The effect of in isopropanol, methanol, ethanol, ether should be shown in the preparation of PEIE:PbX2.
- What is the reason of Jsc difference in PEIE:PbX2 device with various halide components.
- What about active area of cells and the number of devices in the average PCEs?
- How about the stability of devices?
- The adding of integrated Jsc values from EQE are suggested to confirm the results of JV tests.
- The contact angle of PEIE-PbCl2 in Fig. S3 missed.
- The parameters in the test and table should be unified. Such as the Jsc of 8.3 and 8.278 mA/cm2.
Round 2
Reviewer 1 Report
The author have addressed all comments and it can be published in this version